# Identification of Essential Tumor-Infiltrating Immune Cells and Relevant Genes in Left-Sided and Right-Sided Colon Cancers

**DOI:** 10.3390/cancers14194713

**Published:** 2022-09-28

**Authors:** Chen Su, Zeyang Lin, Yongmei Cui, Jian-Chun Cai, Jingjing Hou

**Affiliations:** 1Department of Gastrointestinal Surgery, Zhongshan Hospital of Xiamen University, School of Medicine, Xiamen University, Xiamen 361102, China; 2Institute of Gastrointestinal Oncology, School of Medicine, Xiamen University, Xiamen 361005, China; 3Department of Pathology, Zhongshan Hospital of Xiamen University, School of Medicine, Xiamen University, Xiamen 361102, China; 4Department of Pathology, The Third Affiliated Hospital, Sun Yat-sen University, Guangzhou 510080, China

**Keywords:** MDSC, colorectal cancer, immune infiltration, LCP1, ITGB2, IKZF1

## Abstract

**Simple Summary:**

Differences in oncogenes between left-sided colon cancer and right-sided colon cancer have been reported in-depth. Tumor-infiltrating immune cells and relevant genes between left-sided and right-sided colon cancers are unclear. Bioinformatic analysis was used to identify these hub immune cells and relevant genes. Colon cancer outcomes are associated with changes in MDSC infiltration, and therefore LCP1, ITGB2, and IKZF1 may be novel targets for immunotherapy.

**Abstract:**

Backgrounds: Colorectal cancer is the third most prevalent cancer worldwide. A right-sided colon cancer patient typically has a worse prognosis than one who has a left-sided colon cancer. There is an unclear understanding of how left-sided colon cancer differs from right-sided colon cancer in tumor-infiltrating immune cells (TIICs) and relevant genes. Methods: The Cancer Genome Atlas provided RNA-seq data and clinical information regarding colon adenocarcinoma. We conducted a single-sample gene set enrichment analysis (ssGSEA) to quantify the level of 24 immune cells infiltrating the tissues. Based on an analysis of univariate Cox regression, immune cell types associated with survival were identified. Weighted gene co-expression network analysis (WGCNA) was used to identify hub genes related to location and critical immune cells. Based on the Search Tool for the Retrieval of Interacting Genes (STRING), interaction potential was predicted among the hub genes. Hub genes that influence outcomes through immune infiltration were identified using the least absolute shrinkage and selection operator (LASSO). Then, we used the TISIDB database (a repository portal for tumor–immune system interactions) to validate the correlation between hub genes and immune cell infiltration. Finally, immunohistochemical assays were conducted to determine the levels of proteins expressed by critical TIICs and cancer cells. Results: Colon cancers on the right side of the body had higher levels of myeloid-derived suppressor cells (MDSCs) than on the left side. There were three key genes: LCP1, ITGB2, and IKZF1. It was found that their expression was linked to poor prognosis and an increased level of MDSC infiltration. An immunohistochemical study confirmed these findings. Conclusions: There is a higher rate of MDSC infiltration in right-sided colon cancer when compared with left-sided colon cancer. COAD outcomes are associated with changes in MDSC infiltration, and therefore LCP1, ITGB2, and IKZF1 may be novel targets for immunotherapy.

## 1. Introduction

A leading cause of cancer death worldwide is colorectal cancer, according to the World Health Organization [1]. It has been shown that right-sided colon cancer has a worse prognosis than left-sided colon cancer [2]. Genetic and molecular heterogeneity is likely to account for these differences, which influences immunotherapy [3].

Tumor-infiltrating immune cells (TIICs) are inherent to the tumor microenvironment and play a significant role in tumor progression period, prognosis, and immunotherapy responses [4]. Myeloid-derived suppressor cells (MDSCs) are heterogeneous populations of immature myeloid cells that are identifiable by phenotypic characteristics, such as CD14^+^, CD11b^+^, CD33^+^, and HLA-DR^low/−^. Other markers have also been reported to be expressed by MDSCs in many tumors [5]. The cells of CRC induce immunosuppressive MDSCs, facilitating tumor progression with factors such as TGF-β, arginase, nitric oxide, and reactive oxygen species. It is observed that MDSCs are more prominent in the blood and tumors of patients with CRC compared to healthy subjects, and MDSC levels are correlated with the stage and metastatic status of CRC tumors. The progression-free survival on chemotherapy for patients with a high percentage of MDSCs was also significantly shorter [6]. It has been reported that CRC MDSCs inhibit autologous T cell proliferation in vitro and blocking the MDSC function restores the production of interferon-gamma by T cells [3].

However, differences in TIICs between left-sided and right-sided colon cancer are still poorly understood, despite studies finding that they contribute to cancer progression. As a result, this study compared left- and right-sided colon cancers with respect to the amount of immune cell infiltration.

## 2. Materials and Methods

### 2.1. Gene Expression Profile Data

From The Cancer Genome Atlas, 402 primary tumor and 40 solid tissue normal data were downloaded for RNA-seq analysis. As a criterion for inclusion, complete medical information was required (specifically, the age, gender, pathological phase, TNM stage, point of survival, and time of survival), as well as where the surgery was performed. Among the specimens collected, 120 left-sided COAD tumors, 175 right-sided COAD tumors, and 27 normal solid tissue specimens were included in the dataset.

### 2.2. Cellular Infiltration by Immune Cells

In the R package GSVA, enrichment scores for immune-related terms were determined by comparing TCGA gene expression profile data with the immune cell metagenes set [7]. By integrating the differences between empirical cumulative distribution functions of the gene ranks, the ssGSEA ranks genes according to their absolute expression in a sample. Using a unity distribution, a ssGSEA score was normalized to a score of zero for each immune cell type and a score of one for the maximal immune cell type. It was found that 28 immune cells infiltrated the tumor’s microenvironment. A recent publication [8] was used to compile gene panels for each type of immune cell.

### 2.3. Univariate Cox Regression Analysis

A univariate Cox regression analysis was conducted using R’s survival package. Immune cells associated with outcome were examined using a univariate Cox model. Other risk factors including MSI score, KRAS, NRAS, BRAF, tumor location and TNM stage were also analyzed with a univariate Cox regression model. Calculations were also performed for hazard ratios and their corresponding confidence intervals. A statistically significant difference was defined as one with a *p* value less than 0.05.

### 2.4. WGCNA

The WGCNA, a package for processing gene co-expression networks in R [9], was used to construct the gene co-expression network. First, we selected the highest quartile of gene expression variance. The top 13,780 varying genes from the 295 colonic primary tumor specimens in TCGA, and their standard deviations and relevant documentation, were obtained for further analysis. Outlier samples with connectivity lower than 2.5 were excluded from the clustering tree map. A trait heat map and a sample tree diagram were constructed to illustrate how genes express similar traits. Each gene module was separated into a separate spectrum. As input data for WGCNA, we used gene expression profiles and related information, and, after removing 25 outlier samples, we constructed sample dendrograms and trait heatmaps. The function pickSoftThreshold was used to calculate scale-free topology fitting indices R2 with respect to the soft-thresholding power β. The β value was selected if R2 reached 0.8. A threshold of β = 3 was chosen for this study. Our network was built after we determined the soft threshold. To construct the network, the gene expression matrix was transformed into a topological overlap matrix (TOM), and the degree of dissimilarity was used to determine gene dendrograms and module colors. For module detection, hierarchical clustering was used to produce a dendrogram based on dissTOM for genes by function hclust. In the module, 30 genes are set as the minimum. In our gene dendrogram, we obtained thirteen modules by dynamically shearing the initial module, setting the abline to 0.25, merging the modules with high similarity of features in the gene dendrogram, and using a subset of genes for the abline.

The location and MDSC were included in our analysis. To measure the degree of association between clinical features and the gene module, the Pearson correlation coefficient was calculated between the sample vector of these variables and the characteristic gene of the module, and correlation analysis diagrams were created between the gene module and clinical data. Using the immune cell gene modules most relevant to MDSC, we determined which gene modules were most relevant. The 30 most interconnected genes were selected for further analysis based on the connectivity within the modules.

### 2.5. Identifying Critical Genes via LASSO

LASSO regression was used to narrow the range of target genes (obtained from WGCNA analysis) in order to obtain a more refined model, thereby reducing the number of variable numbers and decreasing overfitting.

### 2.6. Correlation among the Critical Genes, Immune Cell Infiltration, KRAS, NRAS, BRAF, and MSI-Score Using the Corrplot R Package and TISIDB (an Integrated Tumor–Immune System Interaction Repository)

The corrplot R package was used to perform correlation analysis. The correlation among critical gene expression, MDSC, KRAS, NRAS, BRAS, and MSI-score in COAD of TCGA was analyzed. The TISIDB is a web-based platform for analyzing tumor and immune system interactions that integrates heterogeneous data types and analyzes relationships involving target genes and lymphocytes [10]. This study analyzed the correlation between the critical gene expression and MDSC in COAD using TISIDB. Gene expression profiles with critical gene expression were used to infer MDSC abundance using TISIDB. COAD expression scatter plots were analyzed using Spearman’s correlation and statistical significance estimates.

### 2.7. PrognoScan Database Analysis

Using Kaplan–Meier analysis, we explored possible associations between critical genes and CRC survival using the PrognoScan database [11]. Survival of the patients was analyzed based on high and low expression.

### 2.8. Clinical Samples

A set of ethical guidelines was followed, including the Declaration of Helsinki (1975) [12], the International Ethics Standards for Human Biomedical Research [13], and the rules and regulations of the National Natural Science Foundation of China. An international consortium of medical science organizations, including the World Health Organization, supported the study [14]. Research approval was obtained from Xiamen University’s Institutional Medical Ethics Committee. All clinical samples were collected from June 2015 to October 2019 at Zhongshan Hospital of Xiamen University in Xiamen (Fujian Province, China) with the informed consent of patients who underwent surgery. Patients with colon cancer diagnosed by preoperative colonoscopy or pathological biopsy, and who did not receive neoadjuvant chemotherapy or radiotherapy in different locations were included. Sigmoid colon cancer and descending colon cancer were defined as left colon cancer. Ileocecal carcinoma, ascending colon cancer, and transverse colon cancer were defined as right colon cancer. A total of 30 pairs of clinical samples were collected, 15 of which were left-sided samples and 15 of which were right-sided samples (Table 1).

### 2.9. The Immunohistochemistry

Paraffin-embedded and paraffin-fixed tissue sections were deparaffinized in xylenes and rehydrated in alcohols to their original state. A series of antigen retrieval steps was followed by 30 min of blocking at room temperature in avidin/biotin blocking buffer and 3% BSA. In this study, antibodies were used for 60 min at room temperature, followed by two phosphate-buffered saline rinses and diaminobenzidine substrate staining, within the kit (Maixin Biotech, Fuzhou, China, KIT-9710). Hematoxylin was used to counterstain the samples. An Olympus BX51 upright microscope was used for immunohistochemistry images. It was evident from brown staining that the cells were immunoreactive. Antibodies included: Anti-CD11b antibody (1:200, ab133357, Abcam, Cambridge, UK), Anti-CD14 antibody (1:200, ab1883322, Abcam), Anti-CD15 antibody (1:200, ab241552, Abcam), Anti-LCP1 antibody (1:250, 13025-1-AP), Anti-ITGB2 antibody (1:250, 10554-1-AP, Proteinetch, Toyo, Japan), and Anti-IKZF1 antibody (1:250, 12016-1-AP, Proteinetch).

### 2.10. Statistical Analysis

Three independent experiments were conducted to obtain values with mean ± standard deviation. When multiple comparisons were made, the one-way analysis of variance with Bonferroni’s post-test was used, whereas when pair-wise comparisons were made, the Student’s t-test was used. Pearson’s correlation test was used for correlation analyses. Statistically significant differences were defined as those with a *p* value less than 0.05.

## 3. Results

### 3.1. Quantify the Infiltration of Immune Cells

Using ssGSEA, we quantified immune cell infiltration based on mRNA data. The following is a list of 28 infiltrating immune cells: T cells with CD4 and CD8 activation, dendritic cells, natural killer cells with CD56bright, CD4 T cells with central memory, CD8 T cells with effector memory, CD4 T cells with effector memory, CD8 T cells with effector memory, natural killer cells, natural killer T cells, type 1 T helper cells, type 17 T helper cells, CD56dim natural killer cells, dendritic cells in their immature form, macrophages, MDSCs, neutrophils, plasmacytoid dendritic cells, regulatory T cells, type 2 T helper cells, immature B cells, mast cells, memory B cells, monocytes, and T follicular helper cells. (Figure 1A,B).

### 3.2. Relationship between Immune Cells and Prognosis

In order to identify immune cells associated with prognosis, a univariate Cox regression was conducted. Poor prognosis was associated with a hazard ratio (HR) greater than 1, while good prognosis was associated with a hazard ratio (HR) less than 1. The immune cell MDSC (HR more than 1) was an independent risk factor (Figure 2A). BRAF (HR more than 1) was an independent risk factor (Figure 2B).

### 3.3. WGCNA and Critical Module Identification

The data underwent preprocessing and were analyzed by WGCNA in order to determine the modules that contained the strongly correlated genes. After discarding 25 outlier samples, WGCNA was performed on the 13,780 gene expression profiles. The power of β is equal to 3 (scale-free R2 is equal to 0.86) and is used as the soft-threshold to avoid scaling in the network (Figure 3A). The network was built while we determined the soft-threshold. Thirty genes were set as the minimum number in the module. Dynamic tree shearing was used to divide the initial module, setting the abline to 0.25 (Figure 3B). The modules were merged according to their high similarity in the gene cluster dendrogram (Figure 3C), thus obtaining 13 modules.

We analyzed the modules’ correlation between location information, immune cells, and critical genes and created heat maps. A correlation coefficient of 0.89 between the black module and MDSC was found to be the highest (Figure 4). There was a strong correlation between tumor location and MDSC for 30 genes in the black module. The WGCNA algorithm was described in detail by Bin et al. [15].

### 3.4. Identifying Critical Genes Using STRING and LASSO

The black module was preprocessed using STRING and Cytoscape software to construct protein–protein interaction networks [16]. As shown in (Figure 5), proteins are represented by nodes and related interactions by lines. PPI network analysis was used to identify a network of proteins involved in MDSC. These 30 genes were identified using LASSO. Cross-validation was used to determine the adjustment parameter λ (when λ is equal to 0.02250974), so that the error rate was the lowest (Figure 6A,B). We selected LCP1, ITGB2, and IKZF1. Afterwards, survival analyses were performed on each of the three critical genes.

### 3.5. Confirming the Correlation among the Critical Genes, Immune Cell Infiltration KRAS, NRAS, BRAF, and MSI-Score Using the Corrplot R Package and the TISIDB Database

The respective relationship among expressions of three critical genes, MDSC, KRAS, NRAS, BRAF, and MSI-score were analyzed using the corrplot R package (Figure 7A) and TISIDB database. KRAS, NRAS, BRAF, and MSI-score did not show significant correlation with the three critical genes and MDSC (r^2^ < 0.6). LCP1, ITGB2, and IKZF1 expression were significantly positively correlated with MDSC (Figure 7B). This finding suggests that MDSC infiltration is a significant contributor to the expression of LCP1, ITGB2, and IKZF1 within the microenvironment of COAD.

Kaplan–Meier analysis was used to analyze the correlation between expression levels of these critical genes and overall survival in CRC. There was an association between poor survival and high levels of LCP1, ITGB2, and IKZF1 (Figure 7C).

## 4. Immunohistochemical Detection of Immune Cell Types

To measure infiltration levels of MDSC on both sides of COAD, we chose CD11b, CD14, and CD15 as the phenotypic markers. The immunohistochemical analysis showed that CD11b and CD15 exhibited significant differences between left- and right-sided COAD. The relative expression levels of CD11b, CD14, and CD15 were also evaluated. In left and right COAD, CD11b and CD15 expression showed statistically different expression levels (*p* is equal to 0.0282 and *p* is equal to 0.0384, expression), while the expression of CD14 showed no significant difference (Figure 8). These results suggest that MDSC infiltration is more significant in right-sided COAD than left-sided COAD. Through further experimental verification, we revealed that the expression of LCP1, ITGB2, and IKZF1 were significantly upregulated in right-sided colon cancer samples compared to that of left-sided colon cancer samples (*p* < 0.05) (Figure 9). These results showed the positive correlation between MDSC infiltration and the expression of LCP1, ITGB2, and IKZF1 within the microenvironment of COAD, consistent with bioinformatical analysis.

## 5. Discussion

Colon cancer from the right side of the colon contains higher levels of MDSC than cancer from the left side. Our findings and those of other investigators suggest that increased levels of MDSC in colon cancer are significantly correlated with poor outcomes. The present study showed for the first time that right-sided colon cancer tissues express high levels of MDSC, correlating with poor outcomes. These immune infiltration cell differences highlight the potential mechanism of malignancy in this cancer. Immature Myeloid cells accumulate in cancer and are known as MDSCs. They suppress anti-tumor immunity by inhibiting the functions of several immune effector cells, particularly T cells [17]. Tolerance developed between T cells and tumors is what allows tumors to escape. T cell tolerance is associated with tumor-associated MDSC accumulation in animal tumor models and cancer patients. The presence of MDSCs was correlated with robust T cell receptor (TCR) ζ-chain down-regulation among all T cells [18,19], thereby connecting TCR-mediated antigen recognition to diverse signal transduction pathways [20]. There is a well-established connection between MDSC accumulation and immunosuppression in cancer. After curative surgery, MDSCs are found to promote peritoneal metastasis of CRC and are extensively studied as components of the tumor microenvironment [21]. MDSC levels in patients with cancer were higher than in those with premalignant colon polyps, according to Peiwen et al. [22].

The mechanisms underlying the relationship between critical gene expression and MDSC infiltration in COAD remain unknown. Our analysis of WGCNA across TCGA identified essential gene subsets that may affect MDSC infiltration. WGCNA analysis also identified three critical genes that were associated with tumor location. In agreement with the advanced analysis of STRING and LASSO, high expression of critical genes LCP1, ITGB2, and IKZF1 was identified. A correlation was also found between MDSC infiltration and these critical genes’ expression in the TISIDB database. PrognoScan database analysis indicated that overexpression of LCP1, ITGB2, and IKZF1 negatively correlated with COAD patient survival. A functional relationship was found between LCP1, ITGB2, and IKZF1, suggesting that these genes regulate tumor immunology in COAD by regulating MDSC infiltration. Mechanistic studies may be able to shed light on this relationship. The findings also offer a potential target for enhancing immune therapies (converting immunoreactive tumors into immunoresponsive ones).

A critical gene identified as LCP1 was associated with poor outcomes and high MDSC infiltration levels. LCP1 is pathologically relevant for tumor metastasis [23]. ITGB2 (CD18) is the critical subunit of β2 integrin, which actively participates in the immune response. Among our findings in this study was a strong association between ITGB2 levels and poor outcomes. IKZF1 (Ikaros) is a zinc finger transcription factor of the Krüppel family that regulates normal lymphopoiesis and tumorigenesis. The expression of the three proteins positively correlated with MDSC infiltration levels in colon cancer, suggesting their critical roles in the COAD immune microenvironment. It is with regret that KRAS, NRAS, BRAF, and MSI-score did not show significant correlation with the three critical genes and MDSC. In our future work, we will conduct more in-depth studies on how these proteins affect and regulate the immune infiltration and development of colon cancer.

## 6. Conclusions

There is an independent protective effect of MDSC cells on right-sided colon cancer, and they are associated with poor outcomes. Colon cancer outcomes may be affected by three crucial genes that influence MDSC infiltration. Colon cancer treatment may be redirected in this direction based on these findings. It is acknowledged that the scope of this study was limited; therefore, a subsequent study was conducted. That study used data from public exploration, and its results will need to be verified with external data from other medical centers before being translated into clinical practice.

## Figures and Tables

**Figure 1 cancers-14-04713-f001:**
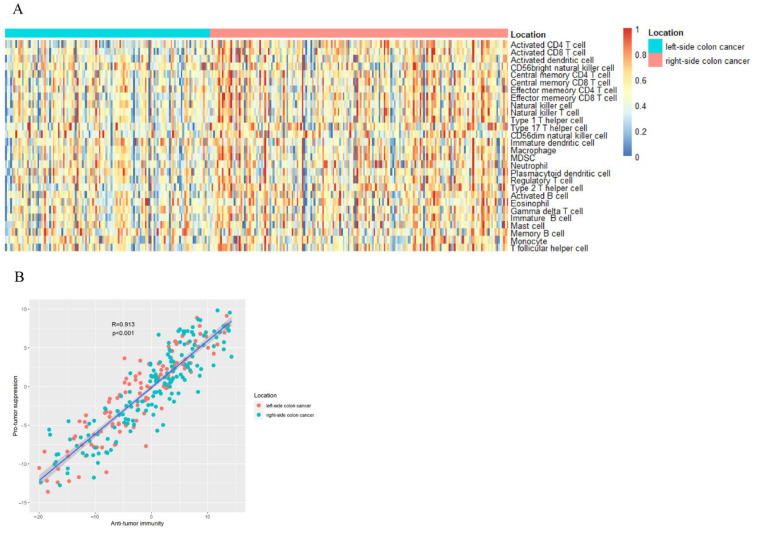
(**A**) An analysis of gene set enrichment using single-sample samples identifies the relative recruitment of immune cells in tumor tissues from different sides of colon cancer. A z-score is calculated based on the relative infiltration of each cell type. (**B**) Relationship between infiltration of anti-tumor immune cells (ActCD4, ActCD8, TcmCD4, TcmCD8, TemCD4, TemCD8, Th1, Th17, ActDC, CD56briNK, NK, and NKT) and their pro-tumor immunosuppressive counterparts (Treg, Th2, CD56dimNK, imDC, TAM, MDSC, neutrophil, and pDC). In Pearson’s correlation, R is the correlation coefficient. Approximately 95% of the shaded area represents a confidence interval.

**Figure 2 cancers-14-04713-f002:**
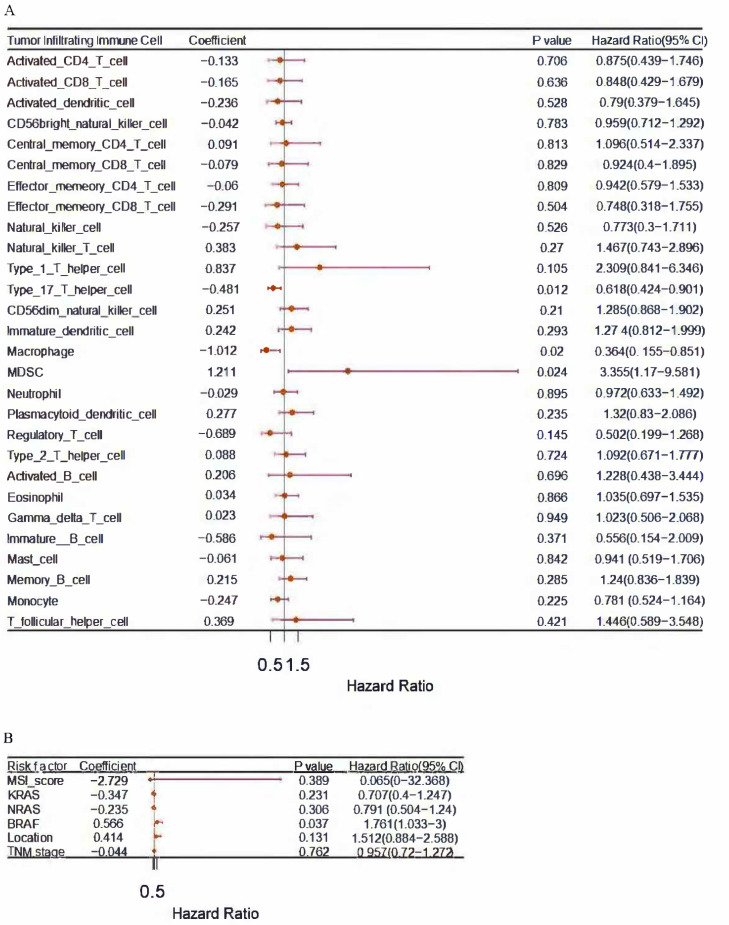
(**A**) Analyzed immune cells from The Cancer Genome Atlas using Univariate Cox regression; (**B**) Analyzed Risk factors including MSI score, KRAS, NRAS, BRAF, tumor location and TNM stage from The Cancer Genome Atlas using Univariate Cox regression; *p* less than 0.05. MDSCs (HR greater than 1) were analyzed, suggesting a negative role for MDSCs in colon cancer.

**Figure 3 cancers-14-04713-f003:**
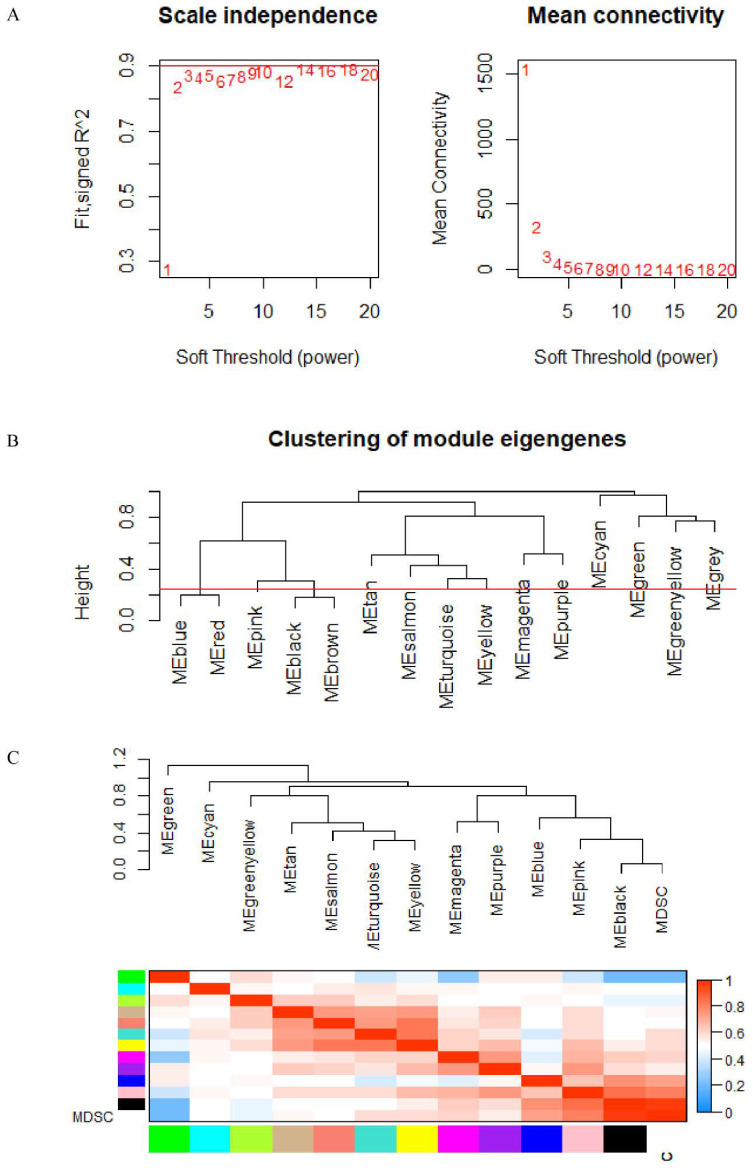
(**A**) Analysis of weighted gene co-expression networks: determination of soft-threshold power. (**B**,**C**) Dendrogram of consensus module eigengenes.

**Figure 4 cancers-14-04713-f004:**
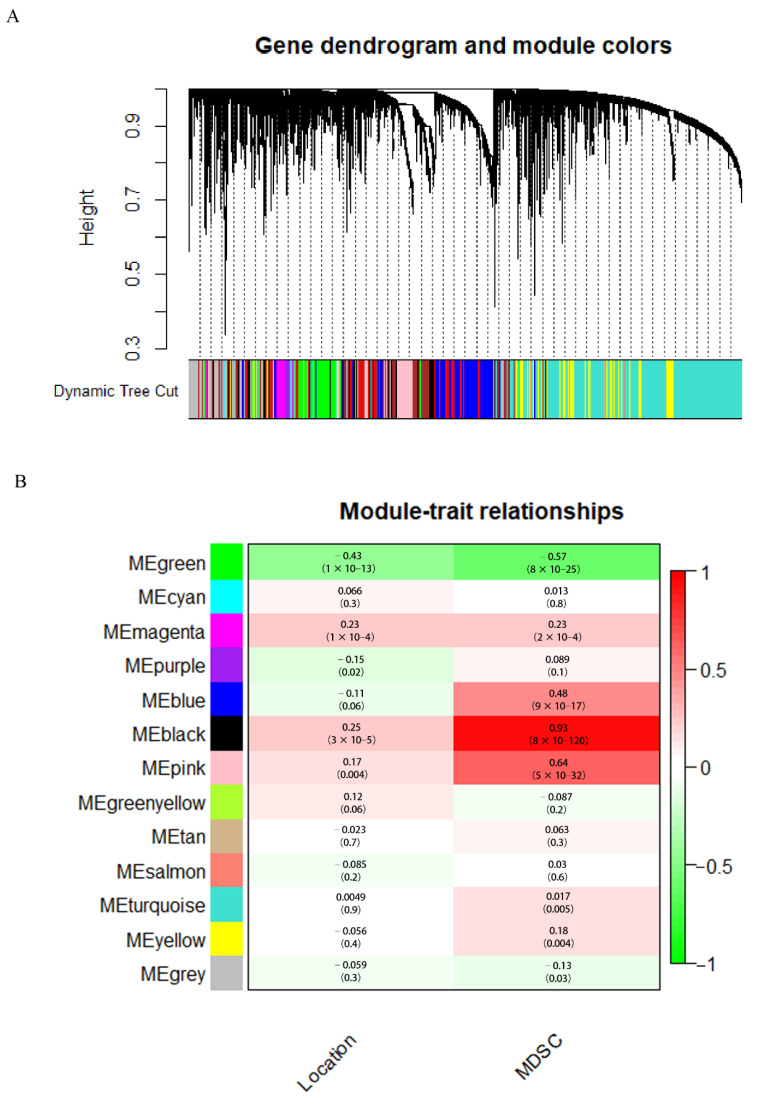
(**A**) Analysis of gene co-expression networks with weighted dendrograms and node color. (**B**) Analysis of gene modules in relation to clinical information.

**Figure 5 cancers-14-04713-f005:**
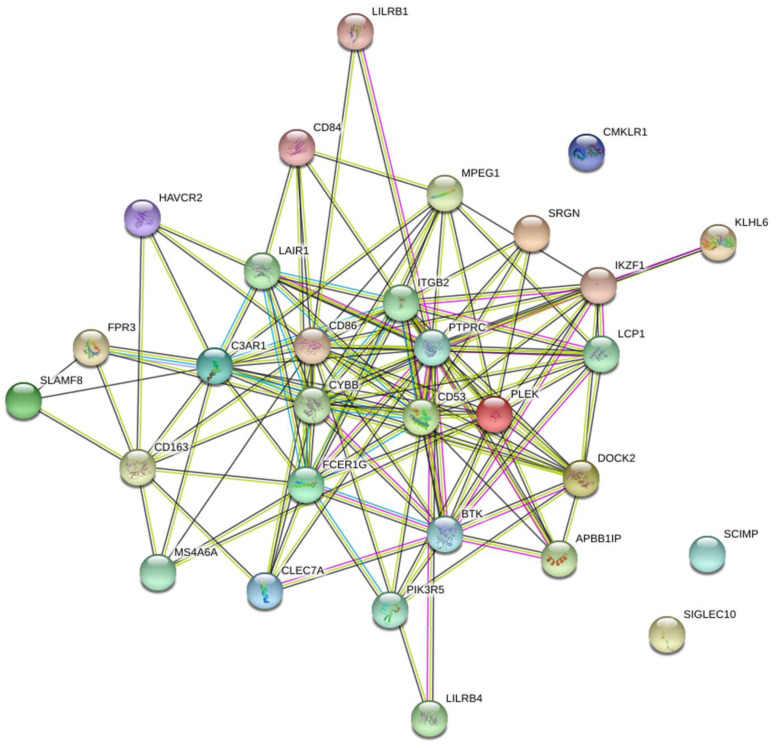
Network of PPI (protein–protein interaction). Genes within the black module were mapped using the PPI network.

**Figure 6 cancers-14-04713-f006:**
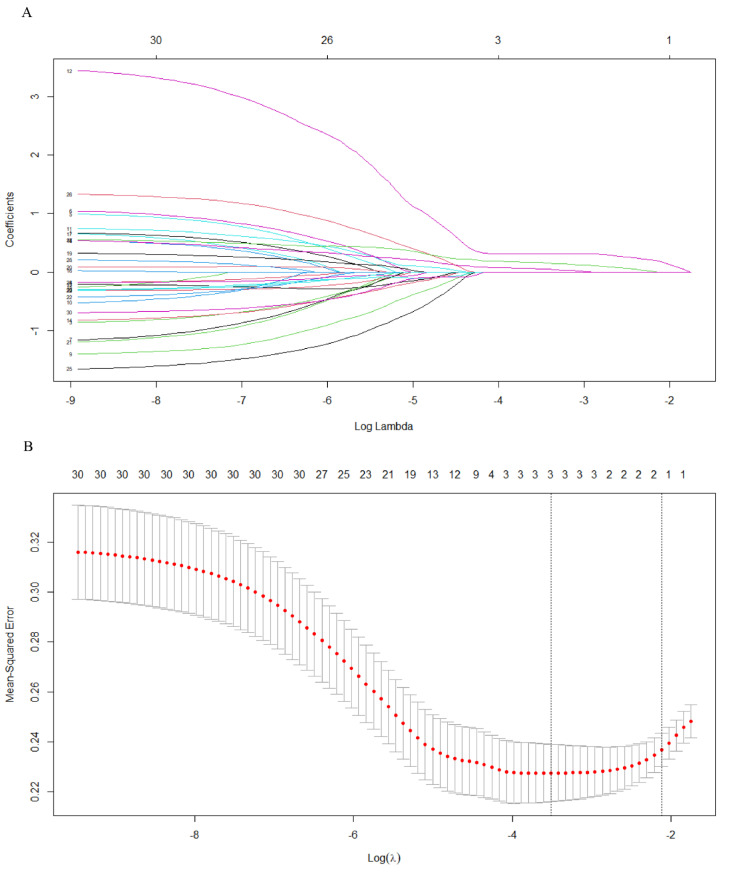
(**A**) The distribution of least absolute shrinkage and selection operator coefficients, biased by partial likelihood. Minimum partial likelihood deviation is indicated by the vertical dashed line. (**B**) LASSO coefficient distribution for 30 genes associated with each other.

**Figure 7 cancers-14-04713-f007:**
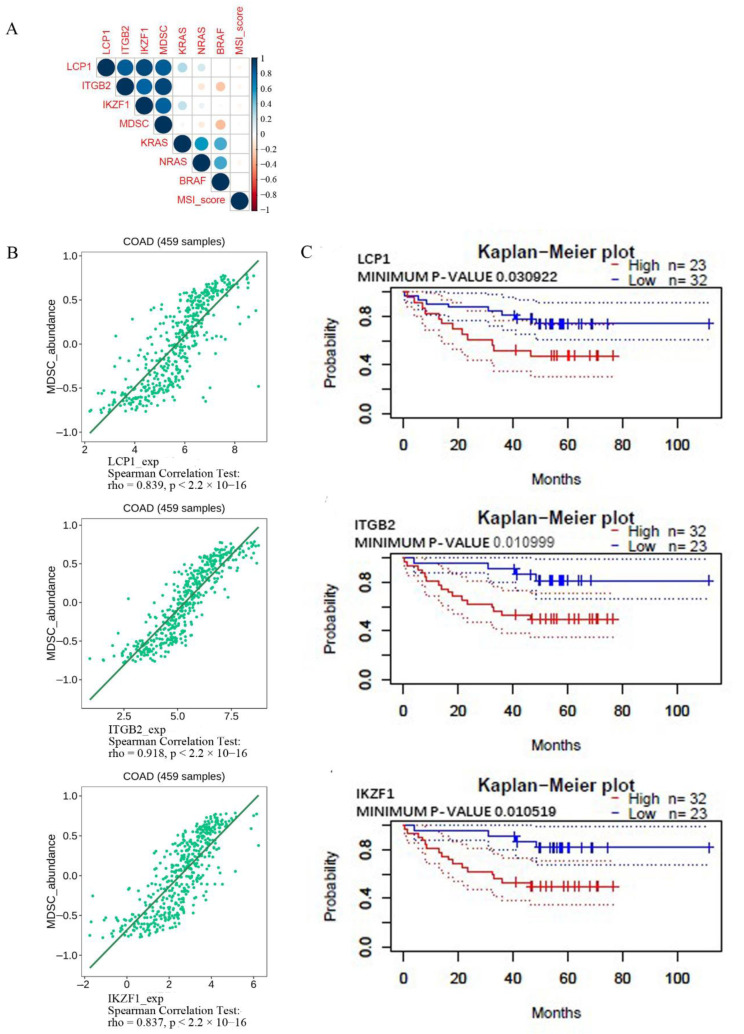
(**A**) The respective correlation among critical gene expression, MDSC, KRAS, NRAS, BRAF and MSI score in COAD of TCGA was analyzed using the corrplot R package. (**B**) Relations between the abundance of myeloid-derived suppressor cells and the expression of LCP1, ITGB2, and IKZF1. The immune-related signatures of MDSC are available on the TISIDB website. (**C**) The overall survival rate. A logrank test was used for analyzing Kaplan–Meier curves stratified by the indicated mRNA levels. Colorectal cancer survival was correlated with genes evaluated in the PrognoScan database.

**Figure 8 cancers-14-04713-f008:**
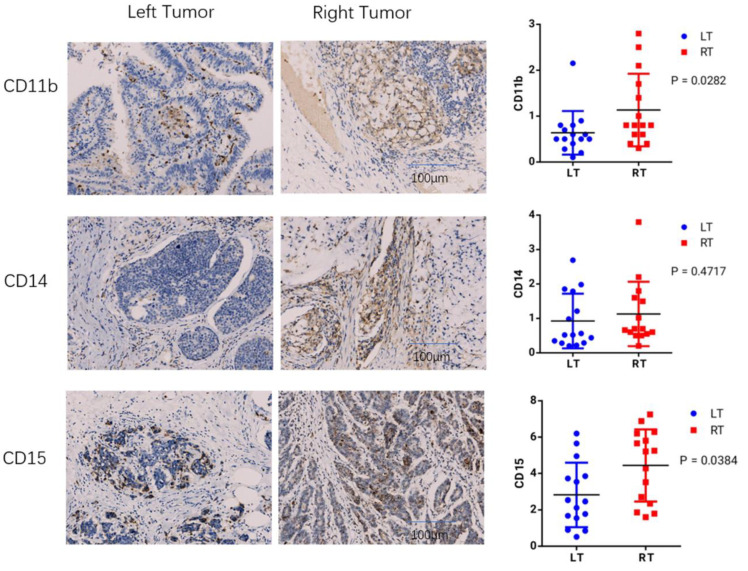
The immunohistochemical analysis. The expression of CD11b, CD14, and CD15 between left- and right-sided COAD were examined by immunohistochemical analysis. The relative expression levels of CD11b, CD14, and CD15 were also evaluated.

**Figure 9 cancers-14-04713-f009:**
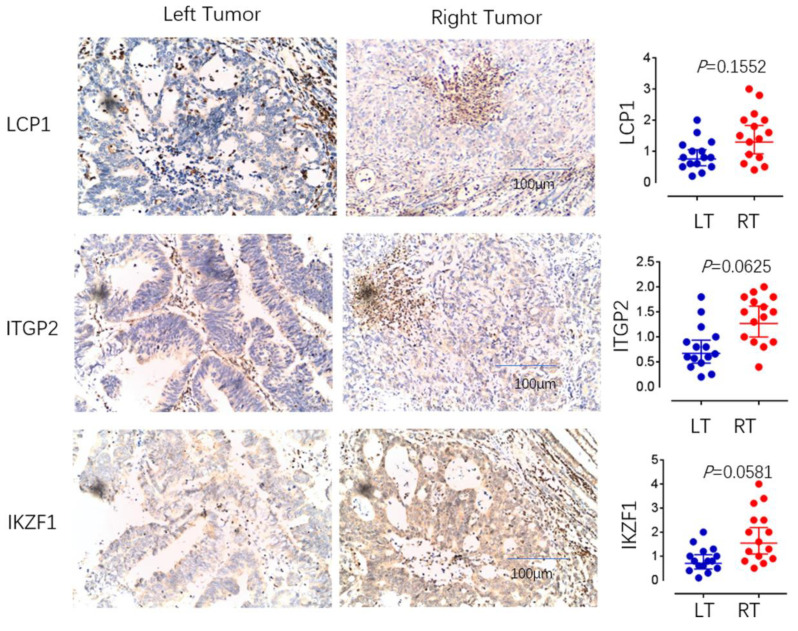
The immunohistochemical analysis. The expression of LCP1, ITGP2, and IKZF1 between left- and right-sided COAD were examined by immunohistochemical analysis. The relative expression levels of LCP1, ITGP2, and IKZF1 were also evaluated.

**Table 1 cancers-14-04713-t001:** Clinicopathological characteristics in colon cancer samples.

Clinicopathological Parameters	Tumor Location		*p*-Value
	Left	Right	
age, years			
≥50	11	12	0.666
<50	4	3	
Sex			
Male	13	10	0.195
Female	2	5	
TNM stage			
Ⅰ + Ⅱ	6	7	0.713
Ⅲ + Ⅳ	9	8	
histology grade			
Well + moderate	10	9	0.705
Poor	5	6	

## Data Availability

Level 3 COAD RNA-seq data were attained as a download from The Cancer Genome Atlas (https://portal.gdc.cancer.gov/). The immune-related signatures are available on the TISIDB website (http://cis.hku.hk/TISIDB/index.php).

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
