# Peer review of "Identification of Essential Tumor-Infiltrating Immune Cells and Relevant Genes in Left-Sided and Right-Sided Colon Cancers"

_cancers, 2022, doi:10.3390/cancers14194713_

Round 1
Reviewer 1 Report
The manuscript entitled "Identification of essential tumor-infiltrating immune cells" is primarily focused on defining the role of tumor-infiltrating immune cells and associated genes to develop a better understanding how the left-sided colon cancer differs from the right sided ones in terms of their prognosis. Overall, the manuscript is well-written with some minor typos or grammatical errors (Introduction: 2nd paragraph - Line 3 - '. [period]' is missing, Line 5- Sentence straucture is wrong; Materials & Methods: Gene expression profile data - Line 1 - Data for the specimen was downloaded not the specimen), hence a through edit is needed. The manuscript is well supported by the data and its appropriate analysis.
Author Response
Dear editor,
Thank you very much for your consideration of our manuscript and request for a revised version. We appreciated your constructive comments .We have gone through the paper and we hoped that we have identified and corrected all the errors.
Again, thank you for taking the time and energy to help us improve the paper.
Reviewer 2 Report
The study conducted the analysis of the level of 24 immune cells infiltrating colon cancer tissues separated as right and left colon cancer. The authors analyzed the genes related with this immune cells. The authors found colon cancers on the right side of the body with higher levels of myeloid-derived suppressor cells than on the left side. In addition they identify LCP1, ITGB2, and IKZF1 as the 3 key genes and their expression was linked to poor prognosis and an increased level of MDSC infiltration.
In this particular study, the methods are well described and sounds and the results are novel.
The paper may be improved:
Material and methods:
Clinical samples section:
The cases were obtained between June 2015 and October 2019 up to a total of 30 (15 each side of the colon. The point is that CRC is so common that there might be a kind of selection for the cases. I would suggest that authors define their selection criteria to avoid any bias that may limit the results of the study.
The sentence: “In addition, we collected matched normal colon mucosae from at least 5 cm away from the tumor to confirm the pathological type of the tumor”, seems to be confuse on how normal colon mucosa can help to confirm the type of tumor…Please clarify.
Table I: It would be nice to add in the analysis of TNM grouped as I-II vs. III-IIV, just as the authors did with histological grade in the same Table..
The immunohistochemistry section:
The authors need to add working dilution of the antibodies in the study.
Author Response
Dear reviewer,
Thank you very much for your consideration of our manuscript and request for a revised version. We appreciated your constructive comments. We have revised our manuscript and Figures. We think that the manuscript has been greatly improved by these revisions and we hope that you will now find it suitable for publication. Our point-by-point responses to comments are detailed on the following pages.Please see the attachment.
We are looking forward to hearing from you at your earliest convenience.
Yours sincerely,
Jingjing Hou

Reviewer 3 Report
Dr. Su and colleagues present their data regarding different immune cell population and gene expression in left vs. right sided colorectal cancer.
Using single-sample gene set enrichment analysis (ssGSEA) of TCGA RNA-seq data, they identified higher levels of myeloid-derived suppressor cells (MDSCs) in reight-sided primary tumors vs. left sided tumors. In addition they identified three different expressed key genes: LCP1, ITGB2 and IKFZ1. The expression of these genes correlated with poor prognosis and increased levels of MDSCs. Results were validated in a small cohort of 30 patients with primary colorectal cancer, which were treated at their center.
The manuscript is well written and the results are very interesting and clinical meaningful.
However, there are some points the authors should address.
11. It is essential to correlated the different expressed genes and immune cell profiles with known clinical and molecular risk factors, especially a correlation with KRAS, NRAF, BRAF and dMMR/MSIhigh status is missing.
22. Figure 2. In univariate correlation analysis the authors identified MDSCs as strong prognostic factor for OS in the CGA cohort. However, patients were included irrespectively of stage and therapy. To validate MDSCs as independent prognostic factor, a multivariate analysis including known risk factors as TNM stage are important and should be added.
33. The validation cohort of 30 patients is very small and patients were included irrespectively of stage. The authors should expand this cohort and focus on Stage II and/or III patients to exclude potential bias.
44. Additional biomarkers as KRAS, BRAF, MSI, NRAS should be added to the validation cohort and should be correlated with MDSCs infiltration and the identified genes.
Author Response

(The authors gave the same response as above.)

Round 2
Reviewer 3 Report
Accept in the present form